# Biomimetic nanocluster photoreceptors for adaptive circular polarization vision

Wei Wen [1,2,4], Guocai Liu [1,2,4], Xiaofang Wei [1,2,4], Haojie Huang[1,2], Chong Wang[2,3], Danlei Zhu[1,2], Jianzhe Sun[1,2], Huijuan Yan[2,3], Xin Huang[1,2], Wenkang Shi[1,2], Xiaojuan Dai[1,2], Jichen Dong[1,2], Lang Jiang [1,2], Yunlong Guo [1,2], Hanlin Wang [1,2] ✉ & Yunqi Liu [1,2] ✉

Nanoclusters with atomically precise structures and discrete energy levels are considered as nanoscale semiconductors for artificial intelligence. However, nanocluster electronic engineering and optoelectronic behavior have remained obscure and unexplored. Hence, we create nanocluster photoreceptors inspired by mantis shrimp visual systems to satisfy the needs of compact but multi-task vision hardware and explore the photo-induced electronic transport. Wafer-scale arrayed photoreceptors are constructed by a nanocluster-conjugated molecule heterostructure. Nanoclusters perform as an in-sensor charge reservoir to tune the conductance levels of artificial photoreceptors by a light valve mechanism. A ligand-assisted charge transfer process takes place at nanocluster interface and it features an integration of spectral-dependent visual adaptation and circular polarization recognition. This approach is further employed for developing concisely structured, multi-task, and compact artificial visual systems and provides valuable guidelines for nanocluster neuromorphic devices.

Artificial vision systems (AVSs) that adapt to complex environments and fulfill multi-task photoperception have become increasingly desired in facial recognition, autonomous vehicles, and visual prostheses. State-of-the-art AVSs are not as exquisite as their biology prototypes in terms of structure simplicity, self-regulation, and multi-functions. For example, photoadaptation devices and neuromorphic phototransistors are designed either with sophisticated multilayers or integration of detectors and processors, increasing manufacturing costs and difficulty[1–8]. It remains a critical challenge to compact a multiple of functions into an all-in-one single cell.

Visual systems of mantis shrimps are equipped with 16 photoreceptors to fulfill multiple tasks of color recognition, adaptive vision, and circularly polarized light (CPL) perception[9]. Though aforementioned functions have discretely been enabled by panchromatic absorbing materials[10], circularly polarized molecule-assemblies[11,12] or chiral compounds[13] and photoadaptive devices. It is of high theoretical and practical value to develop bioinspired hardware that enables parallel processing of color recognition, tunable adaptation, CPL perception, and multi-state readout. Nanoclusters are precise metal atoms coordinated by alternative protective ligands and this unique structure allows tunable physical properties, such as discrete energy levels and sizable bandgaps due to quantum size effects[14]. Additionally, nanoclusters possess excellent photon-to-electron conversion, thus being advantageous for the construction of artificial photoreceptors[15].

Herein, inspired by mantis shrimps, we demonstrate an artificial nanocluster photoreceptor (ACP) array based on a heterostructure formed by chiral nanoclusters and organic semiconductors.

[1]Beijing National Laboratory for Molecular Sciences, CAS Key Laboratory of Organic Solids, Institute of Chemistry, Chinese Academy of Sciences, Beijing 100190, China. [2]School of Chemical Sciences, University of Chinese Academy of Sciences, Beijing 100049, China. [3]CAS Key Laboratory of Molecular Nanostructure and Nanotechnology, CAS Research/Education Center for Excellence in Molecular Sciences, Beijing National Laboratory for Molecular Science, Institute of Chemistry, Chinese Academy of Sciences, Beijing 100190, China. [4]These authors contributed equally: Wei Wen, Guocai Liu, Xiaofang Wei. ✉e-mail: wanghanlin@iccas.ac.cn; liuyq@iccas.ac.cn

Nanoclusters are embedded as in-sensor light valve charge reservoir, enabling decoding wavelength and intensity from incident photons. Furthermore, established nanocluster-conjugated molecule interface (NMI) functions as an adaptative vision by outputting multi-state results with tunable kinetics. On the other hand, by employing the chirality of nanoclusters, ACP perceives circular polarization information. Hence, ACP combines color vision, photoadaptation, and circular polarization vision, thereby opening up opportunities for structurally simplified all-in-one AVSs.

## Results

### Mantis shrimp-inspired nanocluster photoreceptors

Mantis shrimps have incredibly structured eyes comprising thousands of closely packed and parallelly arranged ommatidia. A six-row mid-band (MB) sandwiched by dorsal hemispheres (DH) and ventral hemispheres (VH) is shown in Fig. 1a. A comprehensive illustration of structure-related visual functions in Mantis shrimps is depicted in Fig. 1b and Supplementary Note 1. Ommatidia in the MB region are as optically sensitive units including corneal lens, crystalline cone, and rhabdom (R)[16]. The rhabdom consists of 8 retinular cells with spectral sensitivity from the ultraviolet (UV) to the red[17]. In rows 5 and 6, planes of microvilli in the R8 cells are oriented at an angle of 45° to planes of the R1–R7 cells, acting as 1/2 and 1/4 wave plates to distinguish CPL[18,19]. Moreover, photomechanical changes and tunable filters enable the shrimps to adapt to the variable changes in lighting environments[20,21]. Eventually, optical nerve located at the bottom of each rhabdom transmits signals to the brain.

Inspired by the multi-layered structure of the ommatidia, we fabricate ACP by embedding atom-precise chiral silver nanoclusters beneath a typical organic semiconductor, pentacene layer to form a NMI (Fig. 1c, d and Supplementary Note 2). Chiral Ag nanoclusters with six chiral ligands and 6 chiral Ag cores play a similar role to R8 cells owing to their sensitivity to UV and CPL. Meanwhile, pentacene layer functions analogously to the R1–R7 cells with a broadband photoresponse (Supplementary Fig. 1). Here, pentacene also undertakes the role of optical nerves, transforming incident photons into readable electrical signals (Fig. 1e). Note that the density of photogenerated charge carriers in the pentacene layer can be tuned effectively by trapping and de-trapping dynamics in Ag nanoclusters, under the joint stimulation of light and gate biasing. Therefore, biomimetic visual functionalities, e.g., color vision, adaptive vision, and circular polarization vision, are realized in our all-in-one system (Fig. 1f, Supplementary Note 3). Furthermore, Table 1 illustrates the comparison in terms of photoperception characteristics between our ACP array and counterpart eyes of Mantis shrimps. Moreover, advanced functions are demonstrated in an ACP, manifesting the successful application of our system to multi-task, intelligent devices, and high-density integrated pixelated arrays.

### Microstructure and in-sensor charge reservoir

An ACP array is fabricated on a 4-inch wafer with 200 photoreceptors arbitrarily chosen to test the feasibility of large-scale fabrication (Fig. 2a and Supplementary Fig. 2). Ag nanoclusters and pentacene layers are fabricated by spin-coating and vacuum evaporation, respectively. This combination forms a heterostructure with a smooth and highly-distinguishable interface (Fig. 2b and Supplementary Fig. 3). Detailed fabrication processes are described in Methods. Different means of microscopes are used to elucidate the uniformity of Ag nanocluster layer (Fig. 2c–e and Supplementary Fig. 4). It is discovered that given sufficient nucleation and self-assembly time, Ag nanoclusters form highly crystalline aggregates (Supplementary Fig. 5). However, it represents a challenge for the fabrication of uniform interface with organic semiconductors. We carefully control crystallization and aggregation of Ag nanoclusters by high-speed spin-coating with volatile solvents, hence an ultrasmooth film with a root-mean-square roughness of 0.45 nm is obtained (Supplementary Note 4). No significant X-ray diffraction (XRD) peaks are observed in as-fabricated Ag nanocluster layer, being adequate for homogeneous interface formation in conjunction with pentacene (Supplementary Figs. 6 and 7). Crystallinity of pentacene deposited on Ag nanoclusters is effectively suppressed with respect to that on silica (Supplementary Fig. 7). The hole mobility of pentacene deposited on Ag nanoclusters is measured to be half of the neat pentacene on $SiO_2$ (0.027 $cm^2 V^{-1} s^{-1}$ versus 0.055 $cm^2 V^{-1} s^{-1}$). However, this difference shows limited influence on the biomimetic functions of ACP (Supplementary Fig. 8).

Core-shell Ag nanoclusters underneath pentacene function as charge reservoir to perceive light information, such as spectrum and intensity. Threshold voltage of ACP shifts to the negative/positive direction under gate bias contributing to charges storage/release capacity of Ag nanoclusters (Fig. 2f and Supplementary Fig. 9). Moreover, a counterclockwise hysteresis loop with the threshold voltage ($V_T$) shift of 45 V is observed under illumination (Fig. 2g and Supplementary Fig. 10), which indicates a light-amplified charge trapping and recombination process. Interestingly, photocurrent increases in the positive gate voltage ($V_G$) region while the maximum on-state current decreases as light density increases. This observation demonstrates the synergistic modulation of $V_G$ and light to the magnitude of channel currents. In this way, ACP can display dual-factor stimulated optoelectronic signals. We further quantify ACP's light storage capability by means of hysteresis window, and it is found to be a function of light intensity, wavelength, and $V_G$ scanning range (Fig. 2h and Supplementary Fig. 11). Incident photons in UV and blue spectrum deliver more evident hysteresis than green and red. This result is ascribed to the wide bandgap feature of Ag nanoclusters.

Given the above results, we further study the synergistic effect of gate voltage and light on ACP. Individual application of a positive $V_G$ pulse or a light pulse merely causes a weak increase in current. Nevertheless, concurrent application of positive $V_G$ and light produces a drastic drain current ($I_{DS}$) change from $10^{-8}$ A to $10^{-5}$ A (Fig. 2i and Supplementary Fig. 12). Here, increment in $I_{DS}$ is attributed to Ag nanocluster gating and photoinduced hole accumulation in pentacene. Furthermore, the kinetics of $I_{DS}$ under photoadaptation modes is investigated (Fig. 2j). $I_{DS}$ decreases exponentially with different rates, and it is found to be wavelength-dependent. Adaptation of ACP is written by:

$$I_{adaptation} = A_{adaptation} e^{-t/\tau_{dec}} + I_{initial} \tag{1}$$

$I_{adaptation}$ is the current value of light-exposed ACP, $A_{adaptation}$ is a pre-exponential factor, and $\tau_{dec}$ is time constant. UV and blue photons result in a prompt adaptation or decay in $I_{DS}$ ($\tau_{dec, 365 nm} = 0.25$ s, $\tau_{dec, 460 nm} = 46.4$ s). By comparison, milder declination in $I_{DS}$ levels is produced under green and red light ($\tau_{dec, 525 nm} = 286$ s, $\tau_{dec, 620 nm} = 290$ s). These results are assigned to release of electrons in Ag nanoclusters under excitation of high energy photons, which depletes holes in pentacene by charge transfer and leads to relaxation of $I_{DS}$. As a summary, experiments reveal the spectral-relevant photoadaptation in ACP.

Here, we propose a light-valve model to describe the charge reservoir behavior of Ag nanoclusters (Fig. 2k and Supplementary Note 5). The instant under illumination, accumulation of photoinduced electrons in Ag nanoclusters prompts hole density of pentacene to approach the borderline. After co-modulation of positive $V_G$ and light, ACP switches to closed valve with channel hole density exceeding the borderline and photoinduced electrons preserved in Ag nanoclusters. When the valve is switched on by light, electrons and holes recombine to diminish current levels. Static and dynamic capabilities of charge reservoir are quantitatively analyzed (Supplementary Figs. 13, 14). ACP displays a $10^5$ dynamic range of current levels within 10,000 s and 165 times repeatability in photoresponse and

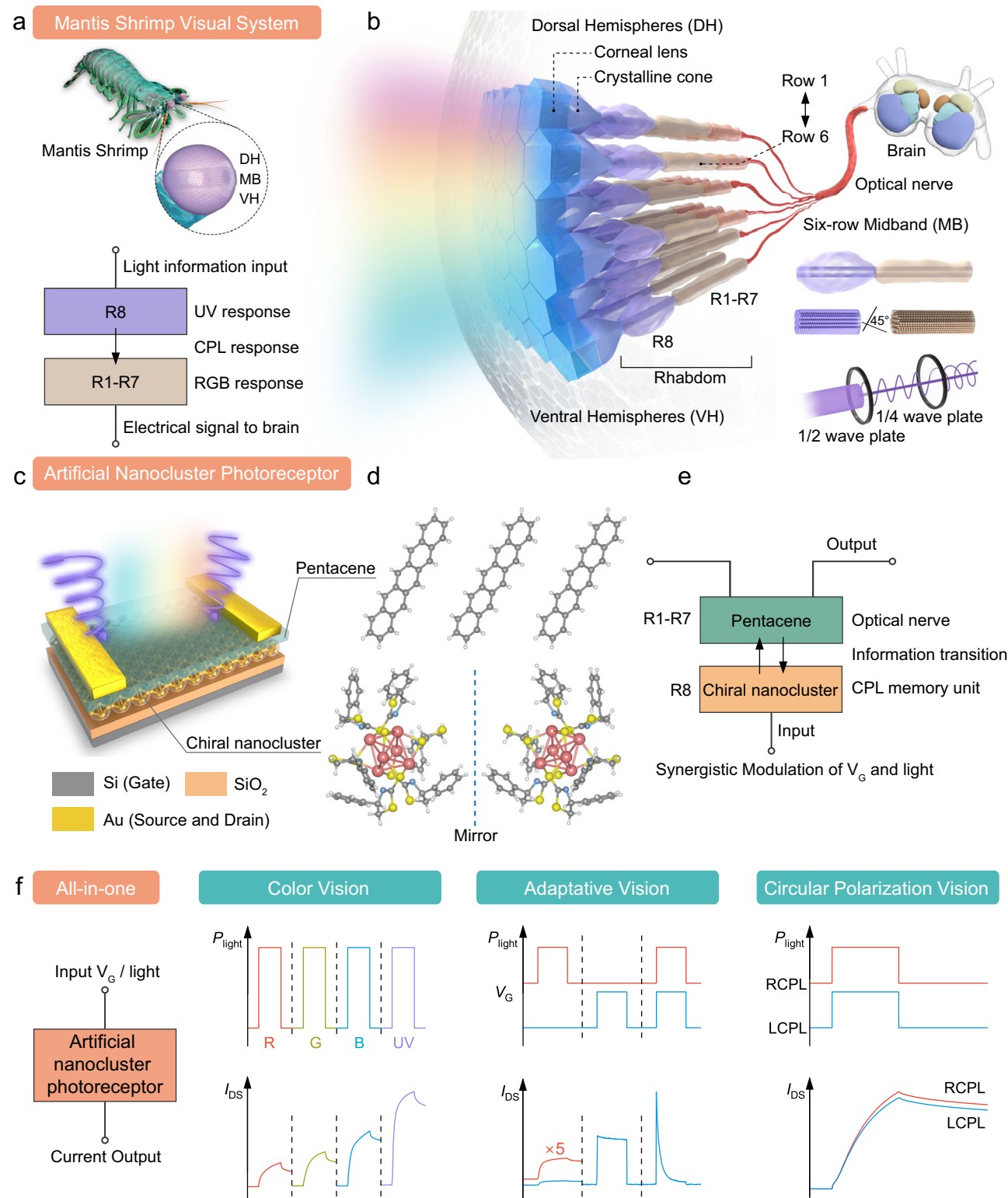

**Fig. 1 | Mantis shrimp-inspired artificial nanocluster photoreceptors. a** A schematic of a mantis shrimp and a front view of its eyes. R1–R8 represent rhabdoms. The eye is comprised of dorsal hemispheres (DH), midband (MB), and ventral hemispheres (VH). **b** Mantis shrimp visual system comprises parallel-arranged ommatidia, optical nerve, and brain. **c** Schematic of an artificial nanocluster photoreceptor (ACP) based on pentacene and chiral silver nanocluster heterostructure sensitive to ultraviolet-visual light and circularly polarized light (CPL). Translucent spheres represent the nanoclusters with 6 chiral silvers (core, small yellow ball) and 6 chiral ligands (shell, translucent section). **d** The chemical structure of pentacene and chiral silver nanocluster enantiomorph. **e** Shrimp-like functions and anatomical structure of ACP. $V_G$ gate voltage. **f** Imitation of all-in-one functional behaviors, including color vision, adaptative vision, and circular polarization vision. $I_{DS}$ drain current, R red, G green, B blue, UV ultraviolet. $P_{light}$ light intensity, LCPL left-handed circularly polarized light, RCPL right-handed circularly polarized light.

**Table 1 | A comprehensive comparison of ACP arrays and mantis shrimp vision system**

| | ACP arrays | Mantis shrimp vision system |
|---|---|---|
| Spectral range | UV, RGB | UV, RGB |
| Layer number | 2 | 8 |
| Thickness | 40 nm | 640 μm |
| Adaptation time | 0.45 s | Hours |
| Perception range | 120 | NA |
| Memory retention time | Over 10,000 s | NA |
| Sites of color vision | Nanoclusters, pentacene | R1–R8; Filters |
| Photoadaptation Mechanism | Nanocluster-conjugated molecule interface | Filters, photomechanical changes |
| Sites of CPL perception | Nanoclusters | R8 |

photoadaptation. In addition, multi-levels in photoresponse can be attained by simply varying the magnitude of $V_G$ impulses without tuning light intensity (Supplementary Fig. 15). Finally, ACP displayed superior shelf life and still retained retention capacity after being deposited for one year (Supplementary Fig. 16). When humidity in testing environments was over 40%, off-state currents in ACP were found to increase remarkably, which might reduce its dynamic range (Supplementary Note 6).

**Charge dynamics in nanocluster-conjugated molecule interface**
Functions of proposed light-valve relies on the interface of pentacene and Ag nanoclusters. Control devices without Ag nanoclusters or with a sandwiched 7 nm tetratetracontane layer into nmI exhibit a faint response and insignificant hysteresis even under white light with intensity of 10 mW cm⁻² (Supplementary Fig. 17). Similar results were observed in parallel devices with interlayer composed of quantum dots and semiconductors (Supplementary Fig. 18 and Supplementary Note 7), which demonstrates indispensable status of our NMI strategy. Interface guarantees a highly efficient charge-transfer process, as proved by time-resolved photoluminescence spectra (Supplementary Fig. 19). The PL lifetime ($\tau_{PL}$) of Ag nanoclusters is evidently shortened from 1.54 μs ($\tau_1$) and 5.74 μs ($\tau_2$) to 0.78 μs ($\tau_1$) and 3.99 μs ($\tau_2$) when interfaced with pentacene. These results demonstrate that photo-generated holes are instantaneously injected into the pentacene channel from Ag nanoclusters.

Ligands are critical for the interfacial charge transfer between Ag nanoclusters and organic semiconductors. Phenyl ligands induce π–π interaction with pentacene and are considered to be responsible for highly efficient charge transfer. To verify this hypothesis, charge density difference is calculated based on a simplified heterostructure model, as shown in Fig. 3a. The phenyl group, together with S atoms and Ag cores forms a direct pathway to capture electrons from pentacene. For comparison, in a control simulation, regions along alkyl ligands that gains and loses electrons are ambiguously identified (Supplementary Fig. 20). Aromatic ligands function as a bridge to enable charge transport through hopping, while alkyl ligands break the coherence of π systems, which impedes charge transport. As a result, alkyl ligand coordinated Ag nanoclusters contained devices have notably inferior performance as ACPs, with their experimentally-derived energy levels and morphology provided in Supplementary Figs. 21, 22. Spin density distribution (SDD) of negatively charged Ag nanoclusters is calculated (Fig. 3b). Lone-electron spin density is completely centralized on Ag core and isolated from pentacene, providing direct evidence for charge reservoir.

To acquire deeper insight into light-valve mechanism of photo-excited interface, we utilize femtosecond-transient absorption spectroscopy to probe the dynamics under excitation of 580 nm and

365 nm lasers. As shown in Fig. 3c and Supplementary Fig. 23, both pentacene and pentacene/Ag nanoclusters produce similar spectral features, wherein pentacene radical cation as a dynamic intermediate produces an excited state peak located at 400−420 nm[22,23]. However, the difference in decay dynamics of excited states is evident. As shown in Supplementary Fig. 24, it takes hole absorption (420 nm) 5 ns to decay to zero (pump: 580 nm) and 30 ps (pump: 365 nm) for penta-cene, being longer than the decay time measured in pentacene/Ag nanoclusters, 1 ns (pump: 580 nm) and 10 ps (pump: 365 nm). To separate the overlapping spectra of the excited states and obtain the unambiguous charge-transfer kinetics, we use the global target ana-lysis to extract the individual component and corresponding time constants (Supplementary Fig. 25). Evolution of three major species is demonstrated by locally singlet excited states (LE), charge-transfer states (CT), and locally triplet excited states (T) in Fig. 3d. Here, the single fission is also considered as a competitive process since the triplet excited states are observed at a femtosecond timescale[24,25]. As shown in Fig. 3e, lifetime of the CT ($\tau_{CT,T}$) for pentacene (74 ps) is nearly six times the length of pentacene/Ag nanoclusters (12 ps) under the 580 nm pump light. $\tau_{CT,T}$ in heterostructure is even shortened to 4.9 ps when 365 nm pump light is used, fully elucidating the role of Ag nanoclusters to accelerate the charge recombination to T state. Hence, photoexcited interface directly exerts influence on holes behavior in pentacene.

Interfacial charge recombination-induced photoadaptation is probed by Kelvin probe force microscopy (KPFM). Surface potential of pentacene decreases with the increment in light intensity, indicating holes depletion under illumination (Fig. 3f). Shorter wavelength leads to more remarkable reduction in surface potential, being consistent with the transient absorption and photoinduced behavior of ACP (Fig. 3g). Therefore, a mechanistic model of ACP is proposed (Fig. 3h). Filled electrons on discrete energy levels will significantly alter the Fermi level of Ag nanoclusters. With low hole density in pentacene, photogenerated excitons are separated at NMI, resulting in an incre-ment of channel hole density. Meanwhile, captured electrons in Ag nanoclusters directly lead to the upshift of Fermi level. When light with a short wavelength or high intensity is applied, a large number of excited electrons prompt the recombination rate at NMI and deplete holes in pentacene, leading to rapid current attenuation. Thus, filled hole/electron levels in pentacene/Ag nanoclusters determine the photoadaptation dynamics. It is worth emphasizing that gate voltage is responsible for charge trapping/de-trapping dynamics rather than the polarity of photogenerated carriers in Ag nanoclusters. To conclude, we can facilely modulate ACP through light and gate impulses.

**Spectral-dependent visual adaptation**
In response to the brightness changes in environments, visual adap-tation has evolved as a self-regulated activity in various living crea-tures. The aim of adaptation is to modulate the perception range of eyes to attain the best contrast and clear images. The magnitude of visual adaptation can be modulated by both incident light wavelengths and luminance intensities. As far as ACP, photoadaptation functions as a reservoir to regulate the channel current, keeping it at an appropriate level. ACP shows stepwise strengthened, electric-mode adaptation with increased $V_G$ (Fig. 4a). This is ascribed to the gate-driven hole transfer from pentacene to Ag nanoclusters. Furthermore, the opera-tion mode of either photopic adaptation or scotopic adaptation is tunable and dependent on $V_G$ applied (Fig. 4b and Supplementary Fig. 26). We define adaptation index (AI) by $I_t/I_i$ to evaluate the mag-nitude of light adaptation under various wavelengths. ACP has a stronger adaptability to short-wavelength and high-intensity light. It is explained by photons with high energy being capable of inducing charge accumulation in wide bandgap nanoclusters (Fig. 4c). Percep-tion range (PR) and adaptation time ($T_{adapt}$) are critical parameters to quantitatively analyze the light-induced adaptability from the device.

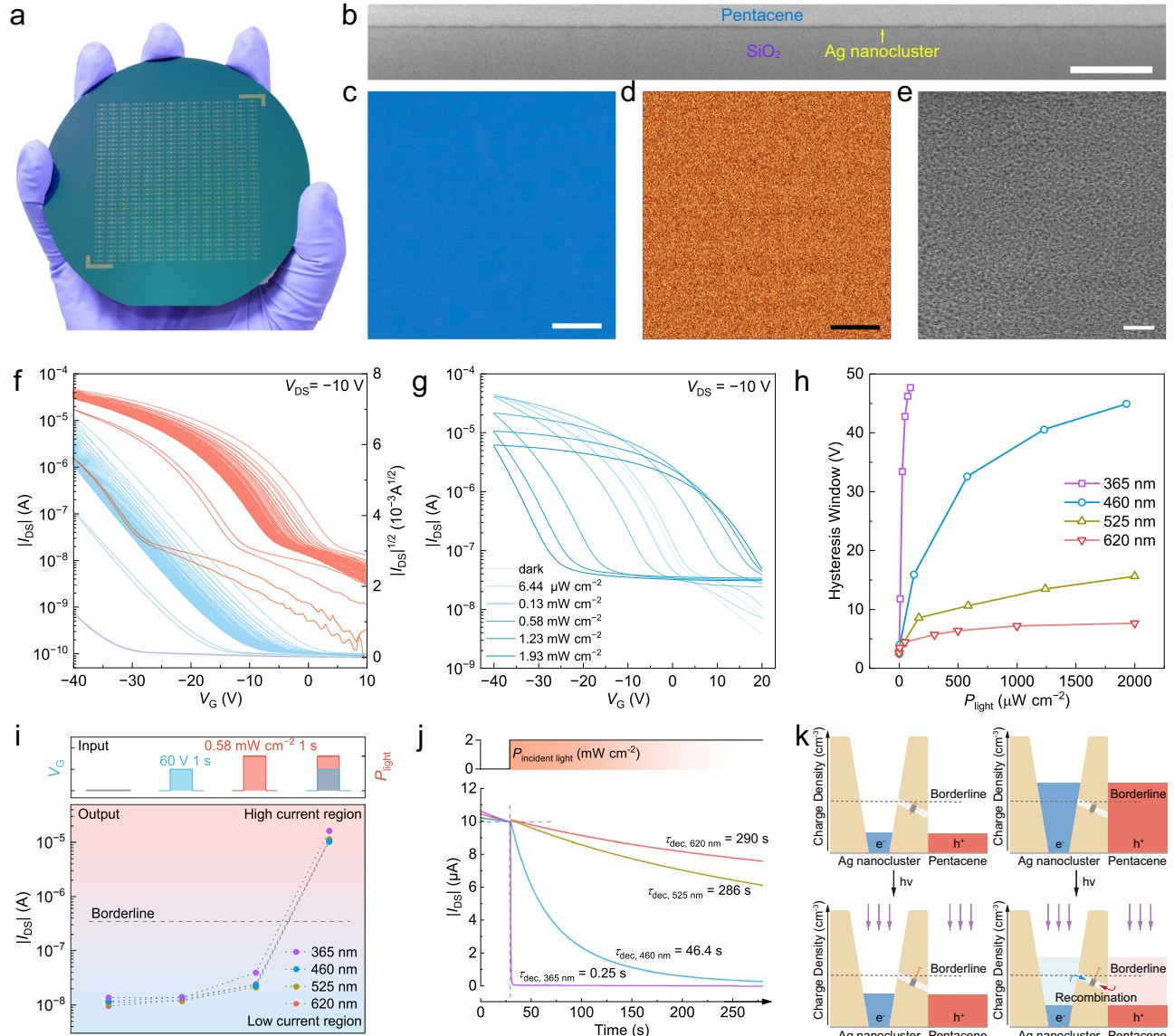

**Fig. 2 | Microstructure and in-sensor charge reservoir. a** Optical image of a wafer-scale artificial nanocluster photoreceptor (ACP) array. Channel width and length of the device are 4500 μm and 20 μm, respectively. **b** Cross-sectional transmission electron microscope image of a pentacene/Ag nanoclusters/SiO₂ stack. Scale bar: 200 nm. **c** Optical microscope image of Ag nanocluster film. Scale bar: 8 μm. **d** Scanning near-field optical microscopy of Ag nanocluster film. Scale bar: 2 μm. **e** Cryo-transmission electron microscope image of Ag nanocluster film. Scale bar: 20 nm. **f** Transfer characteristics of a typical device under 30 times of repetitive bidirectional scanning or after $V_G$ pulse (−60 V or −80 V, 1 s) under darkness. **g** Hysteric transfer characteristics window in ACP towards different light intensity (460 nm). **h** Hysteresis window in a device as a function of incident light intensity with different wavelengths (UV: 365 nm, blue: 460 nm, green: 525 nm, red: 620 nm). **i** Synergistic effect of gate voltage and light on ACP. **j** Real-time photoadaptation under incident light (2 mW cm⁻²) with a multiple of wavelengths ($V_{DS}$ = −10 V, $V_G$ = 0 V). **k** The diagram of light-valve model and charge reservoir of Ag nanoclusters.

PR is defined by:

$$PR = 20 \times \log\left[\frac{I_{max}}{I_{min}}\right] \quad (2)$$

Adaptation time is quantified by the time required for current to decrease from $I_{max}$ to $I_{adapt}$ or adaptation speed. For ACP, $I_{max}/I_{min}$ can reach $10^6$ and $PR$ is calculated to be 120. We extract the $I_{DS}$ as a function of light intensity ($P_{light}$) under variable $V_G$ as shown in Fig. 4d. High photon density significantly leads to more prompt declination of device current under the condition that negative $V_G$ is applied. Hence, under $V_G$ = −30 V, adaptation time of ACP to UV light (365 nm, 0.8 mW cm⁻²) can be shortened to 0.45 s (Supplementary Fig. 27).

Accuracy is given by $(I_{max} - I_{adapt})/(I_{max} - I_{min})$ and it reaches 99.75% under this condition. As shown in Fig. 4e and Supplementary Table 2, ACP presents a high $PR$ and the shortest $T_{adapt}$ compared with existing state-of-the-art adaptable devices[1–3,26–34], thereby proving the successful adoption of nanoclusters for fast adaptive vision sensors.

To demonstrate spectral-dependent adaptation, we fabricate a 5 × 5 ACP pixel array to perceive patterns and record the image immediately after light illumination (Fig. 4f and Supplementary Fig. 28). Initially, the photocurrents of ACPs are all in a photobleached state after stimulation ($V_G$ = 60 V, 1 s; light impulse, 2 mW cm⁻², 1 s) and this represents the scene where human eyes could not obtain any information. After a 25 s interval under illumination with various wavelengths, ACP array can distinguish letters in different colors by means of variable current levels. Hence, ACP array demonstrates its

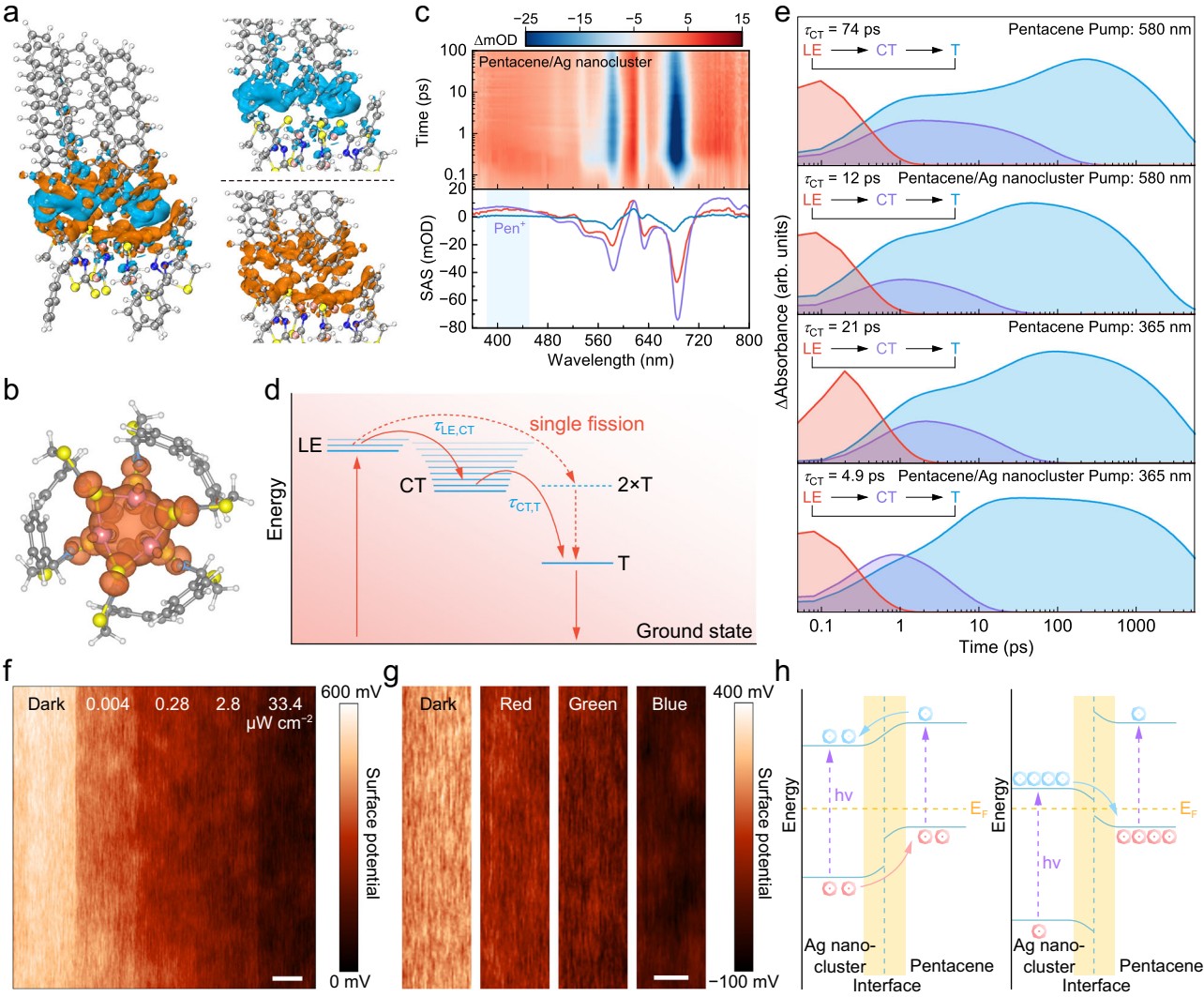

**Fig. 3 | Charge dynamics in nanocluster-conjugated molecule interface.**
**a** Charge density difference of pentacene/Ag nanocluster heterostructure. Blue and orange colors indicate the negative (electrons decrease) and positive (electrons increase) values, respectively. The isosurface value is 0.002 e Å$^{-3}$. **b** Spin density distributions (SDD) of negatively charged Ag nanoclusters. **c** Top: femtosecond-transient absorption spectroscopy of pentacene (80 nm)/Ag nanocluster (20 nm) film recorded with several time delays between 0.01 ps and 100 ps in the visible region ($\lambda_{pump}$ = 580 nm) at room temperature. Pen$^+$: pentacene radical cation. Bottom: species-associated spectra (SAS) of the film above from global and target analysis. Orange line: LE state, green line: CT state, blue line: T state. **d** Schematic diagram of the process in the film through localized excited states (LE), charge-transfer state (CT), and triplet state (T). $\tau_{LE,CT}$ and $\tau_{CT,T}$ are the lifetimes of excitons. **e** Evolution of the population of the LE, CT, and T states. **f** Surface potential of pentacene/Ag nanocluster film under different intensities of light generated by 473 nm lasers. Scale bar: 0.5 μm. **g** Comparison of the surface potential of pentacene/Ag nanocluster film under different wavelength light (473 nm, 532 nm, 633 nm). **h** Energy-band diagram of the heterostructure under light illumination. Left dissociation mode, Right recombination mode, Blue ball electron, Red ball hole.

capacity for object recognition with spatial and color information by producing adaptive signals. Furthermore, emulation of retinal damage under UV light has been performed. After 20 s illumination of UV light, $I_D$ regains its off-state value and discharges all information previously perceived.

## Circular polarization vision
Animals encounter the risk of information disclosure by predators, prey and competitors during intraspecific communication by postures, sound waves, etc[18]. Polarization of light is an appropriate medium for encoded information exchange and potential high-frame event-triggered vision[35–37]. Mantis shrimps have evolved circular polarization vision as an encrypted way to convey information. Circular polarization vision, capable of detecting and analyzing CPL, has aroused significant attention due to its potential applications in chiral sensing and information encryption[38]. Although state-of-the-art CPL

detectors have achieved recognition of CPL, lack of memory for circularly polarization information limits their potential in AVSs. Core-shell chiral Ag nanoclusters endow ACP with CPL storage capabilities. Figure 5a shows the circular dichroism spectrum of Ag nanocluster film, the maximum dissymmetry factor ($g_{CD}$) value for Ag nanoclusters has been measured to be ~$\pm 1.3 \times 10^{-3}$ at 277 nm.

By the chiral assembly of Ag nanoclusters, circularly polarized vision, including response, identification, adaptation, and memory to CPL, can be realized in a single biomimetic cell. Distinct saturated photocurrent signals of ACP exposed to CPL illumination are shown in Fig. 5b, c and Supplementary Fig. 29. ACP with S-Ag nanoclusters has higher photocurrent under right-handed circularly polarized light (RCPL) than left-handed circularly polarized light (LCPL). By comparison, opposite phenomenon is observed for ACP with R-Ag nanoclusters. When ACP is exposed to the light, the photocurrent increases gradually. As the light is switched off, the current decreases to a steady

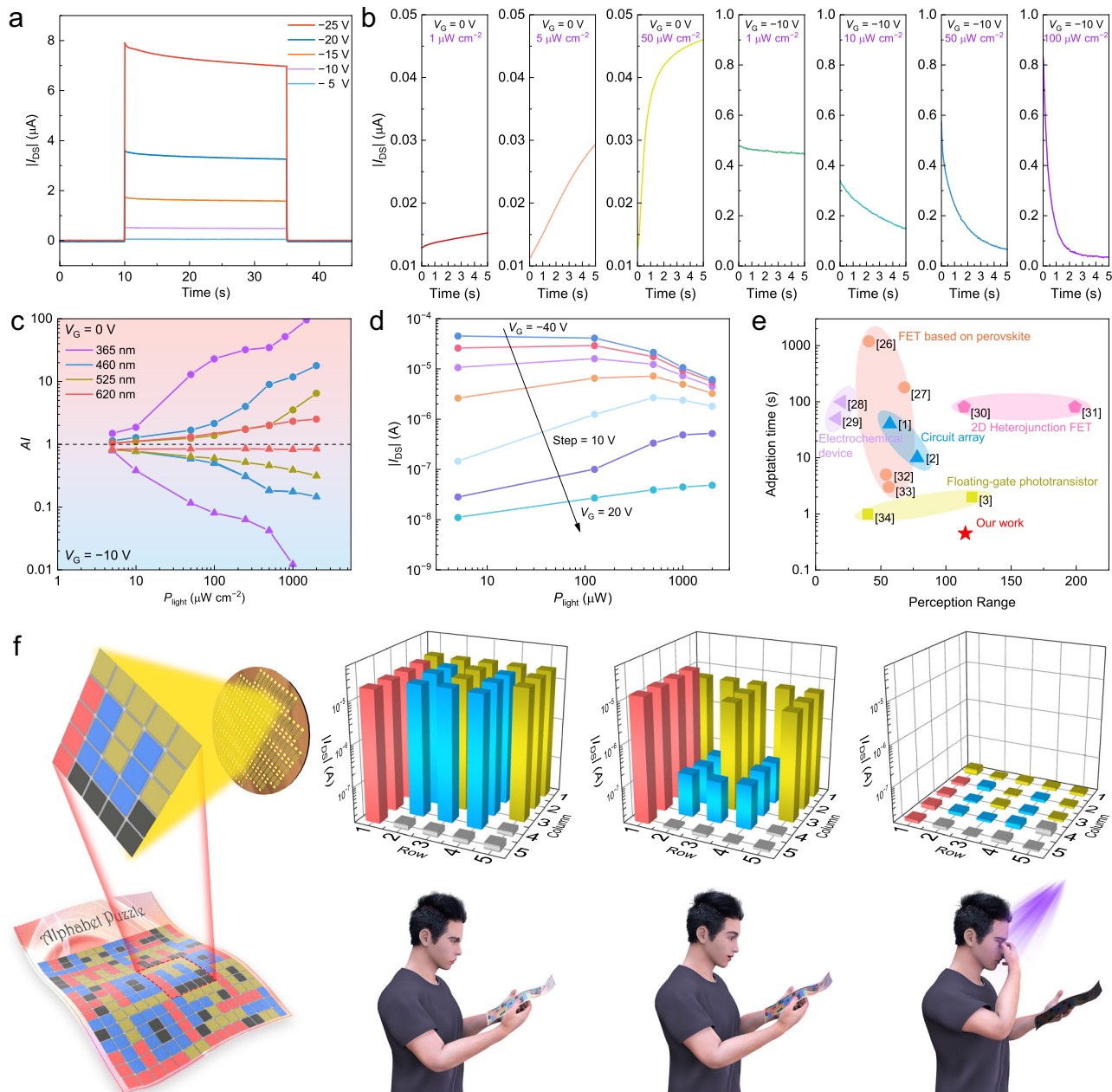

**Fig. 4 | Spectral-dependent visual adaptation. a** Real-time current recording under various $V_G$ amplitude. **b** Scotopic adaptation ($V_G = 0$ V) and photopic adaptation ($V_G = -10$ V) under different $P_{light}$ values. **c** $AI$ values measured as a function of $P_{light}$, $V_G$ values and incident photon wavelengths. $AI$ is defined as: $I_t/I_i$, $I_t$ and $I_i$ are defined as the device current instantaneously read from the end and the beginning of light impulse. **d** $I_{DS}$ plotted against $P_{light}$ under various $PR$ biasing conditions.

**e** Comparison of the perception range ($PR$) and adaptation time ($T_{adapt}$) of adaptable device in this work with results in literature. **f** Schematic of the spectral-dependent photopic adaptation. Mapping of signals in 5 × 5 pixel array, which perceives the pattern of letters I (620 nm), F (525 nm), U (460 nm) from a colored alphabet that is illuminated by a multiple of wavelengths.

state, demonstrating a memory operation that has not been reported in other CPL detectors. ACP can store and distinguish RCPL/LCPL currents, as shown in Fig. 5d and Supplementary Fig. 30. After 15 s RCPL/LCPL impulses, current switches into an exponential decay curve. Hence, it is reasonable to speculate that ACP can still maintain and distinguish CPL after retention of 100,000 seconds. The reproducibility of CPL discrimination capability has been verified in Fig. 5e. Moreover, adaptation of ACP to CPL light with low light density can be achieved. $AI$ reaches 0.19 while $T_{adapt}$ is prolonged to a timescale of 7 s. Under alternative switching of RCPL and LCPL, ACP clearly perceives the polarization of incident CPL with good reproducibility (Fig. 5f).

## Discussion

We report biomimetic nanocluster photoreceptors based on visually adaptable chiral-nanocluster-conjugated molecule interface. Light-assisted, tunable Femi energy levels of nanoclusters function as electron reservoir and they precisely control trapping and release of charge carriers in proposed artificial photoreceptors. Inspired by mantis shrimps, visual adaptation in ACP is accomplished by the modulation of photogenerated carriers at the interface of organic molecules and Ag nanoclusters. Wavelength and intensity-dependent adaptations are further demonstrated by the recognition of shapes and colors. Taking advantage of the chirality of Ag nanoclusters, ACP is enabled to distinguish CPL light. As a result, circular polarization vision

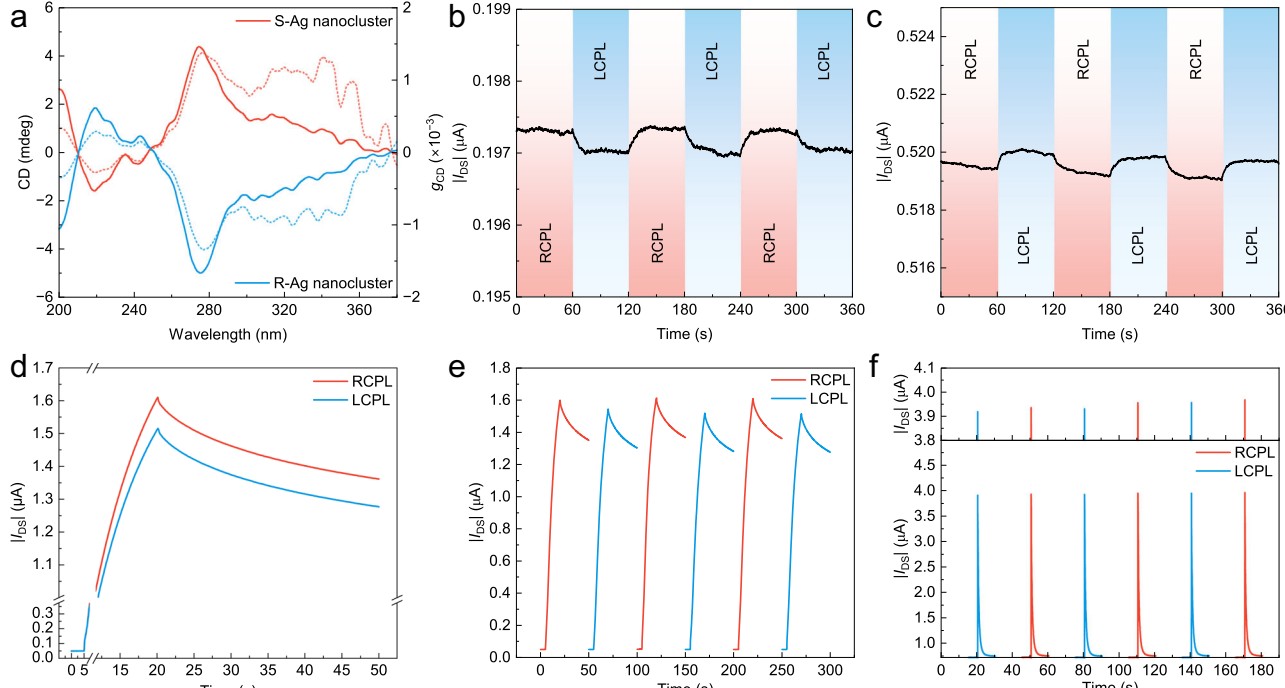

**Fig. 5 | Circular polarization vision. a** Circular dichroism (CD) spectroscopy and $g_{CD}$ characteristics of the chiral Ag nanocluster film on quartz substrates. Solid line: CD value. Dotted lines: $g_{CD}$ value. Real-time saturated drain current in an artificial nanocluster interfaced photoreceptor based on S-Ag nanoclusters (**b**) and R-Ag nanoclusters (**c**) under alternatively switched left-handed circularly polarized light (LCPL) and right-handed circularly polarized light (RCPL). Illumination conditions: 270 nm, 100 µW cm⁻². Biasing conditions: $V_G = 0$ V, $V_{DS} = -1$ V. **d** Real-time $I_{DS}$ of an artificial nanocluster interfaced photoreceptor device based on S-Ag nanoclusters under and after 15 s LCPL and RCPL illumination (270 nm, 100 µW cm⁻²), $V_G = 0$ V, $V_{DS} = -5$ V. **e** Repetitive $I_{DS}$ current level in response to alternatively switched LCPL and RCPL illumination. **f** Circular polarization visual adaptation based on S-Ag nanoclusters. Illumination conditions: 270 nm, 100 µW cm⁻². Biasing conditions: $V_G = 0$ V, $V_{DS} = -10$ V, $V_G$ pulse is set as $-30$ V.

is successfully emulated and highly similar to the functions of mantis shrimps. To further explore the light-valve mechanism of Ag nanoclusters, charge storage site and charge-transfer pathway in NMI are surveyed by transient absorption and photoluminescence techniques. In summary, ACP presents an emulation of the complex visual system in mantis shrimps. Its multi-task feature integrates panchromatic adaptation and circularly polarized vision through a simple architecture. As a perspective, this bioinspired system can be upgraded to meet the needs of vision-related AI hardware and communication of encrypted information.

Nanocluster-embedded artificial photoreceptors construct a concise model to explore optoelectronic behavior of nanoclusters. We reveal charge carrier dynamics and photoinduced electron transport in the nanocluster-organic heterostructure interface. The core-shell structure of nanoclusters enables them with charge reservoir ability and protective ligands function as signal transduction pathways linking nanoclusters and organic materials. In doing this, we further develop wafer-scale fabrication technology of nanocluster electronics and lead nanoclusters into the era of artificial intelligence.

## Methods

### Materials
(R/S)-4-Isopropylthiazolidine-2-thione and (R/S)-4-Phenylthiazolidine-2-thione were purchased from Sigma-Aldrich as ligands of nanoclusters. Silver nitrate was purchased from Macklin. Pentacene was purchased from Tokyo Chemical Industry Co., Ltd. (TCI). Tetratetracontane was purchased from Sigma-Aldrich.

### Preparation of chiral silver nanoclusters
Preparation of Ag nanoclusters is conducted in similar protocols to the literature[39]. Silver nitrate (1 mmol) and ligand (1 mmol) were dissolved

in a mixture of 1 mL DMAc/CH₃CN ($V$ : $V$ = 3:1) to form a yellowish solution with fluorescence. Afterwards, the yellowish block Ag nanocluster crystals were obtained by slow-evaporation of solvents in darkness at room temperature for 1 day.

### Fabrication of ACPs
P-type silicon wafers with thermally oxidized SiO₂ (~300 nm thick) were used as substrates. As the cleaning procedures, wafers were soaked into the piranha solution (70 vol% H₂SO₄ and 30 vol% H₂O₂) for 10 min, then rinsed successively via ultrasonication in de-ionized water, acetone, and ethanol for 5 min, respectively. After being blown dry with nitrogen gas, wafers were further passivated by oxygen plasma for 10 min. Chiral Ag nanocluster crystals were dissolved in tetrahydrofuran (5 mg mL⁻¹) and spin-coated on the wafer with a speed of 6000 rpm for 30 s. Due to its low boiling point (66 °C), most tetrahydrofuran volatilized during spin-coating. Under a pressure of $6 × 10^{-4}$ Pa, 30-nm-thick pentacene was thermally evaporated on the Ag nanoclusters at a rate of 0.1 Å s⁻¹. Finally, 30-nm-thick gold (Au) was thermally evaporated through a shadow mask to define source and drain electrodes. Channel width-to-length ratio was designed to be 225 ($W/L$ = 4500 µm/20 µm).

### Characterization of ACPs
Electrical measurement of all devices was characterized by a semiconductor parameter analyzer (Keithley 4200 A SCS) and source meters (Keysight B2912A). Atomic force microscopy was performed using a NanoMan VS system. SEM images were taken with a Hitachi S-4800 field emission scanning electron microscope. KPFM images were obtained with a NTEGRA Spectra (NT-MDT, Russia) in a Kelvin mode by using a gold-coated silicon probe with force constant ~17 N m⁻¹. The scan rate is 0.5 Hz, and the light was provided by 473 nm,

532 nm, 633 nm lasers. Scan near-filed optical microscopy (SNOM) images were obtained by neaSCOPE. The wavelength of the laser was set to 1331 nm according to the FT-IR spectrum of the Ag nanoclusters. Cryo-transmission electron microscope (Cryo-TEM) image of Ag nanocluster film was captured by Themis 300. The cross-sectional image of the device was captured by transmission electron microscopy (TEM; JEM-2100F, JEOL) operated at 200 kV. The grazing-incidence wide-angle X-ray scattering (GIWAXS) data were obtained at 1W1A Diffuse X-ray Scattering Station, Beijing Synchrotron Radiation Facility (BSRF-1W1A). Steady-state UV-vis spectra were recorded with Perkin-Elmer LAMBDA 1050 spectrometer. Steady-state PL spectra were measured with a Horiba FluoroMax+ spectrometer. Time-resolved photoluminescence (TRPL) spectra were measured on an FLS980 Steady-State and Transient Fluorescence spectrophotometer with an excitation wavelength of 400 nm. HOMO energy levels of nanoclusters were measured by Ultraviolet photoelectron spectroscopy (Kratos Axis Ultra Dld). The XRD data were measured by X-ray Powder diffractometer (Malvern PANalytical Empyrean). CPL was generated by 270 nm LED through a half-wave plate (Thorlabs, WP25M-UB) and a quarter-wave plate (Thorlabs, AQWP05M-340). The intensity of LCPL and RCPL was calibrated by a standard Si detector (Newport, 818-SL/DB).

## Transient absorption measurements

TA spectra were measured by Vitara T-Legend Elite-TOPAS-Helios-EOS-Omni. Light pulses were provided from a Ti: Sa amplified laser system (Legend Elite-1K-HE). Wavelengths of pump light were 365 nm and 580 nm, while the probe from 380 nm to 800 nm was generated by the laser beam onto a $CaF_2$ plate. Global and target analysis was performed by global target analysis, GloTarAn, based on the R-package TIMP[40,41]. Singular value decomposition, decay-associated spectra, and species-associated spectra are obtained by global and target analysis.

## Computational methods

Quantum chemical calculation was carried out using Gaussian 09 software package[42]. The ground-state geometry optimization was performed by Perdew-Burke-Ernzerhof functional using 6–31 g* basis set for H, C, N, and S atoms and Lanl2TZ effective core potentials for Ag atoms[43–45]. The single-crystal structure from CCDC was chosen as the initial guess for ground-state geometry optimization. Afterwards, analyzing wavefunction by Multiwfn[46], charge density difference, and SDD were performed by B3LYP using def2-TZVP basis set for all atoms and visualized with Visualization for Electronic and Structural Analysis[47].

# Data availability

The data are provided to support the plots within this manuscript, and other findings of this study are available from the corresponding authors upon request. Source data are provided with this paper.

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

## Acknowledgements

This work is supported by grants from the Strategic Priority Research Program of the Chinese Academy of Sciences (XDB0520000), the National Natural Science Foundation of China (no. U22A6002, 61890941, 61890943), the National Key R&D Program of China (grant no. 2018YFA0703202), and the CAS-Croucher Scheme for Joint Laboratories (no. CAS20903). A portion of this work is based on the data obtained at BSRF-1W1A. The authors gratefully acknowledge the cooperation of the beamline scientists at the BSRF-1W1A beamline.

## Author contributions

Y.L. conceived the project. W.W., H.W. and Y.L. designed the project and experiments. W.W. and G.L. performed the device fabrication. X.W. performed the spectral characterization. W.W. performed the device characterization. W.W., G.L. and X.W. analyzed the data. H.H. and J.D. performed the theoretical calculation. C.W. analyzed the transient absorption spectra. D.Z. and L.J. provided assistance in the CPL characterization. J.S. provided assistance in SNOM tests. H.Y. performed the KPFM measurements. X.H., W.S., X.D. and Y.G. provided assistance in material characterization. W.W. and H.W. wrote the manuscript. All authors discussed the results and commented on the manuscript.

## Competing interests

The authors declare no competing interests.
