## [Peer Review File · Nature Communications]

Biomimetic Nanocluster Photoreceptors for Adaptative Circular Polarization VisionREVIEWER COMMENTS

Reviewer #1 (Remarks to the Author):

In this manuscript, Wen and co-workers demonstrated an all-in-one artificial photoreceptor array based on chiral nanoclusters. Complex functions including photoadaptation and circular polarized light vision were successfully integrated and demonstrated in this manuscript. The light-valve model proposed by the authors clarifies well the role of nanoclusters in the realization of adaptative optoelectronic behaviors. As-fabricated nanocluster-conjugated molecule interface is remarkably modulated yielding spectrally-dependent photoadaptation and it identifies circularly polarized light. Overall, the proof-of-concept, data and characterization presented are of high quality and focused on timely interdisciplinary questions. In particular, this work should appeal researchers working on nanocluster electronics and neuromorphic device engineering. The manuscript is also well illustrated and written, thus it is accessible to the broadest interdisciplinary readers of this journal. I find that this work matches the highest standards of Nature Communications thus I recommend its publication once the minor points below are suitably addressed.

1) Nanoclusters were functioned as chiral units and charge reservoir due to their chiral ligands and core-shell structure. I am wondering if such nanoclusters can be substituted by carbon dots, perovskite dots, and metal chalcogenides (MCs) dots. Please justify your answer.

2) Two kinds of nanoclusters have been presented, with the alkyl ligand protected nanoclusters presenting inferior performance. Why ?

3) Are the light-valve model and charge reservoir model applicable for all kinds of nanoclusters? Are the Ag nanoclusters in this work able to produce similar photoadaptation when they are used in series with other organic semiconductors other than pentacene?

4) gCD value for nanoclusters is relative lower than other reported chiral materials. Why ?

Reviewer #2 (Remarks to the Author):

Q1. In the proposed ACP device, the synergistic modulation effect of gate bias and light stimulation is of paramount importance in influencing the photo response. Do mantis shrimps also demonstrate variations in their photo response in accordance to external stimuli other than light? Or is there similar situation in R8 that the capacity of Ag NCs in ACP device is modulated by external stimuli?

Q2. In Fig. 4c, the case of $V_g=0V$ with light increases the amount of the current due to the hole accumulation in the pentacene channel. Then, why does the current decrease in the case of negative V_g ? The gate bias seems to only control a dynamics charge trapping of photo-generated carriers into Ag NCs, not the polarity of photo-created carriers.

Q3. What is the definition of adaptation function? Please describe the adaptation in both cases of the shrimp and in the proposed device.

Q4. The chiral structure of Ag NCs plays a crucial role in ACP devices, but no peaks were

detected in both TEM and XRD results. An evidence (ex. XRD, or TEM diffraction pattern) is needed to confirm the chiral Ag NC crystal structure.

Q5. When evaluating the electrical characteristics of a transistor with the channel exposed, it is highly susceptible to external environmental influences. Thus, it is needed a comments on the measurement conditions. For instance, were the measurements taken in a vacuum or in ambient atmospheric conditions? Are there any differences in the retention results depending on the environment?

Q6. In Supplementary Fig. 8, examining the hysteresis loop under different wavelength, it is evident that selectivity is insufficient at lower light intensities depending on the RGB wavelength. What accounts for this phenomenon?

Q7. In Supplementary Fig. 11, under the conditions of $V_G=60$ V and 460 nm illumination at 2 mW cm⁻², stable retention was observed even at a timescale of 10,000 s. However, in Supplementary Fig. 13, the multi-level retention only extends to a timescale of 600 s despite the bias duration being raised to 5 s. Why is this the case?

Q8. (minor) The full abbreviation of LCPR, RCPR is not mentioned at any point

Reviewer #3 (Remarks to the Author):

Stomatopods, commonly known as mantis shrimp have one of the most complex visual systems known in the animal kingdom, with 12 spectral receptors and the ability to perceive linearly and circularly polarized light.

I find the work original even if I don't understand everything, especially the materials part, it's hard to follow since it's not my area of expertise.

I haven't found an article close to this present one, by the same authors.

The most related works on this subject are

Altaqui, A., Sen, P., Schrickx, H., Rech, J., Lee, J. W., Escuti, M., ... & Kudenov, M. (2021). Mantis shrimp-inspired organic photodetector for simultaneous hyperspectral and polarimetric imaging. *Science Advances*, 7(10), eabe3196.

<https://www.science.org/doi/full/10.1126/sciadv.abe3196>

Authors doesn't quote it. The materials used are not the same.

I would ask to quote

Haessig, G., Joubert, D., Haque, J., Chen, Y., Milde, M., Delbruck, T., & Gruev, V. (2021). Bio-inspired polarization event camera. arXiv preprint arXiv:2112.01933.

<https://arxiv.org/pdf/2112.01933>

and

Garcia, M., Davis, T., Blair, S., Cui, N., & Gruev, V. (2018). Bioinspired polarization imager with high dynamic range. *Optica*, 5(10), 1240-1246.

<https://opg.optica.org/optica/fulltext.cfm?uri=optica-5-10-1240>

and

D. Floreano, R. Pericet-Camara, S. Viollet, F. Ruffier et al. (2013)

Miniature curved artificial compound eyes

Proceedings of National Academy of Sciences of USA, PNAS, 2013 Jun 4, 110(23):9267-72

<https://doi.org/10.1073/pnas.1219068110>

The size of the text in the figures is really small everywhere.

Figure 4c explains their position in relation to the state of the art, but it's very small.

For me, it's the most important figure, but the choice of colors isn't very legible, and the font size isn't correct.

There's a lot in their paper in terms of figures, Authors should move more to the Supp info to lighten the paper and make the figures bigger.

Most importantly, the authors should relax the biomimetic aspect with respect to a full mantis shrimp eye as the mantis shrimp eye has 12 spectral receptors and the ability to perceive linearly and circularly polarized light.

I cannot see a comparison in terms of optical characteristics with the Mantis Shrimp natural counterpart eye, as done in the table 1 of Floreano et al. PNAS 2023

Finally, it would have been nice to see a picture or a conceptual view of the final device that would include the electronic read out. I would be very helpful.

A point-by-point response to the reviewers' comments

Comments and Responses for Reviewer #1.

Comments:

In this manuscript, Wen and co-workers demonstrated an all-in-one artificial photoreceptor array based on chiral nanoclusters. Complex functions including photoadaptation and circular polarized light vision were successfully integrated and demonstrated in this manuscript. The light-valve model proposed by the authors clarifies well the role of nanoclusters in the realization of adaptative optoelectronic behaviors. As-fabricated nanocluster-conjugated molecule interface is remarkably modulated yielding spectrally-dependent photoadaptation and it identifies circularly polarized light. Overall, the proof-of-concept, data and characterization presented are of high quality and focused on timely interdisciplinary questions. In particular, this work should appeal researchers working on nanocluster electronics and neuromorphic device engineering. The manuscript is also well illustrated and written, thus it is accessible to the broadest interdisciplinary readers of this journal. I find that this work matches the highest standards of Nature Communications thus I recommend its publication once the minor points below are suitably addressed.

Response:

Thank you very much for your positive opinions and beneficial advice on this work. Under your instructions, we have revised the manuscript with all the changes and updated results marked in red in this version.

Comments 1:

Nanoclusters were functioned as chiral units and charge reservoir due to their chiral ligands and core-shell structure. I am wondering if such nanoclusters can be substituted by carbon dots, perovskite dots, and metal chalcogenides (MCs) dots. Please justify your answer.

Response:

Thank you for your comments. We have noticed that various nanoscale dots such as carbon dots and PbS dots have been reported with their charge-storage capacities and applications thereof in floating-gate memories and some memristors. Similar to nanoclusters, these core-shell dots

mentioned above can feasibly be enabled with chirality by rational design, such as self-assembly and asymmetric ligands. Although it is possible to substitute nanoclusters by these dots as far as functionalization is concerned, nanoclusters still be indispensable theoretically and experimentally due to the following reasons.

Theoretically, the major differences between nanoclusters and nanodots are the sizes, cores and ligands. Compared with known nanodots, nanoclusters are more likely to produce thin film with low roughness due to their smaller physical size (several nm versus tens of nm), which presents an evident advantage for subsequent deposition of semiconductor films with high quality and low defects. Hence, we have selected well-dispersed nanoclusters with a diameter of nearly 1 nm and attained a large stable hysteresis under luminescence. Noble metal nanoclusters are featured by atomic centers (Au, Ag, Cu, Pt) with empty d orbitals. Such a property is anticipated to accommodate plenty of electrons under photophysical processes. Hence, cores of nanoclusters have higher charge-storage capacities than carbon dots, perovskite dots, and metal chalcogenides (MCs) dots as electron reservoir species. Another distinction is the role ligands play in nanoclusters and nanodots. Ligands are primarily adopted as colloidal stabilizers and dispersibility reagents in nanodots. In contrary, ligands directly influence energy levels of nanoclusters due their smaller physical size, which facilitates the charge transfer from protective ligands to the metal cores and thus enabling potential optoelectronic functions. In our work, we find ligands of nanoclusters function as a bridge for dissociation of electrons and holes. A ‘Ligand-assisted charge transfer process’ in nanocluster/organic interface has been elucidated by theoretical calculations and femtosecond transient absorption. As a brief summary, nanoclusters can hardly be substituted by carbon dots, perovskite dots, and metal chalcogenides (MCs) dots in terms of photoadaptive devices. However, we believe nanodots can be rationally designed and synthesized to achieve similar photoadaptive capabilities and they require extensive investigation from peer researchers.

To perform supplementary experiments, we have selected three common commercial quantum dots, carbon dots, PbS dots to substitute nanoclusters and tested their electric characteristics. The aqueous carbon dots were purchased from Suzhou Xingshuo Nanotech Co., Ltd and dispersed in methanol with a concentration of 10 mg mL⁻¹. The oleic acid-capped PbS QDs were purchased from Suzhou Xingshuo Nanotech Co., Ltd and dispersed in toluene with a concentration of 25 mg mL⁻¹. Both of them were diluted to a concentration of 5 mg mL⁻¹. Parallel devices were fabricated with similar

procedures in Ag nanocluster photoreceptors. As a result, these devices substituted by carbon dots and PbS dots demonstrate inferior optoelectronic performances under luminescence, as shown in Supplementary Fig. 18.

Supplementary Fig. 18 | Electrical properties and morphology of quantum-dot-based Ag NCs devices. (a) AFM of PbS dot film with a root-mean-square roughness (R_{RMS}) of 4.34 nm. (b) AFM of carbon dot film with a R_{RMS} of 0.248 nm. (c) Bidirectional transfer characteristics of PbS dots-based photoreceptors. (d) Bidirectional transfer characteristics of carbon dots-based photoreceptors.

The above modifications and additions are attached to **Supplementary information on Page S19 and S44-45**. We also revise the first paragraph in the section named ‘Charge dynamics in nanocluster-conjugated molecule interface’ **in the main text on Page 5** as follows. ‘Control devices without Ag NCs or with a sandwiched 7 nm tetratetracontane layer into NMI exhibit a faint response and insignificant hysteresis even under white light with intensity of 10 mW cm^{-2} (Supplementary Fig. 17). Similar results were observed in parallel devices with interlayer composed of quantum dots and semiconductors (Supplementary Fig. 18 and Supplementary Note 7), which demonstrates indispensable status of our NMI strategy.’

Comments 2:

Two kinds of nanoclusters have been presented, with the alkyl ligand protected nanoclusters

presenting inferior performance. Why ?

Response:

Thank you for your comments. The modifications and additions are attached to **Supplementary information on Page S22-23.**

The aim of presenting two types of nanoclusters is to clarify the structure-performance correlation in ligands. To our knowledge, few research articles have probed the charge-transfer characteristics at the nanocluster/organic semiconductor interface, let alone the role ligands played in signal transduction. It is widely acknowledged that charge transport along co-facially stacked π planes present a critical pathway in organic semiconductors. Hence, it can be inferred that aromatic ligands other than alkyl ligands have higher structure affinity to conjugated backbones in organic semiconductors, enabling more efficient charge transport at the ligand/organic semiconductor interface and potentially lower injection barrier. Meanwhile, the less 'conductive' alkyl chain would lead to electron tunneling rather than hopping that is readily available in aromatic ligands. Furthermore, we have proved our speculation by experiments and theoretical calculation. The alkyl ligand protected nanoclusters have notably inferior performance in memory capacity and retention time, being due to the following reasons:

Aromatic ligands function as a percolation pathway to transport charges, instead alkyl ligands break the coherence of π systems. (ii) We find that two kinds of nanoclusters have slight difference in energy levels, which in turn governs the interfacial injection barrier or efficiency. The highest occupied molecule orbitals (HOMO) of nanoclusters are estimated from ultraviolet photoelectron spectroscopy (UPS) and the lowest un-occupied molecular orbitals (LUMO) are calculated from optical band gap, as shown in Supplementary Fig. 18 and Supplementary Table 3. (iii) Ligand-induced morphology evolution of nanoclusters. We find that the root-mean-square roughness of the alkyl ligand protected nanocluster film is 13.2 nm, being much higher than the aromatic one, which is responsible for non-uniformity, interfacial defect formation with high density and deteriorated performance.

On Page S24, we have appended:

Supplementary Fig. 22 | UPS spectra of (a) aromatic ligand protected nanoclusters and (b) alkyl ligand protected nanoclusters films. UV-vis spectra of (a) aromatic ligand protected nanoclusters and (b) alkyl ligand protected nanoclusters films.

Supplementary Table 2 | Summary of energy levels of Ag nanoclusters.

	Aromatic ligand protected	alkyl ligand protected
HOMO (eV)	-5.79	-5.34
LUMO (eV)	-3.06	-2.56
E_g (eV)	2.73	2.78

On Page S22, we have appended:

Supplementary Fig. 21d | AFM of alkyl ligand protected nanocluster film.

The above modifications and additions are attached to **Supplementary information Page S30-S33**. We also revise the second paragraph in the section named ‘Charge dynamics in nanocluster-conjugated molecule interface’ **in the main text on page 5-6** as follows. ‘Aromatic ligands function as a bridge to enable charge transport through hopping, while alkyl ligands break the coherence of π systems which impedes charge transport. As a result, alkyl ligand coordinated Ag NCs contained devices have notably inferior performance as ACPs, with their experimentally-derived energy levels and morphology provided in Supplementary Fig. 21 and 22.’

Comments 3:

Are the light-valve model and charge reservoir model applicable for all kinds of nanoclusters? Are the Ag nanoclusters in this work able to produce similar photoadaptation when they are used in series with other organic semiconductors other than pentacene?

Response:

We thank you for your insightful question. The modifications and additions are attached to **Supplementary information on Page S41-42**.

Supplementary Note 5: In-sensor light valve charge reservoir model

Each material has its unique characteristics, hence the light-valve model and charge reservoir model are not universal in all kinds of nanoclusters. In fact, we propose the light-valve model and charge reservoir model to account for the photoadaptation behavior. In our work, Ag NCs are revealed with adaptative response to light, charge capture, and tunable Fermi energy level, hence electronic states of Ag NCs can be synergistically tuned by gate bias and light. These characteristics are the prerequisite for the construction of nanocluster-type photoreceptors. To address your concern, there are four requirements to achieve a successful nanocluster/organic interface. (i) Nanoclusters should

be composed of precious metal cores that offer plenty of empty orbitals for extra electrons and function as charge reservoir. (ii) Aromatic ligands are required to form effective hopping pathway for charge carrier transition to π -conjugated backbones of semiconductors. (iii) Nanoclusters should be synthesized with high absorption coefficient. (iv) π -conjugated backbone of semiconductors should be sterically adjacent to the ligands of nanoclusters to ensure effective charge transfer. In a control experiment, we use an interlayer of tetratetracontane (TTC) to physically isolate the nanocluster and the pentacene layer, leading to suppressed photoresponse in the device. According to these rules, not every semiconductor satisfies the requirements. Organic semiconductors with long and bulky alkyl side chains are probably not suitable for photoadaptation.

To verify our inference, we substituted pentacene by C8-BTBT as the channel semiconductor. Under a pressure of 6×10^{-4} Pa, 30-nm-thick C8-BTBT was thermally evaporated on the Ag NCs with a rate of 0.1 \AA s^{-1} . Then, 30-nm-thick gold (Au) was thermally evaporated through a shadow mask to define source and drain electrodes. Channel width-to-length ratio was designed to be 225 ($W/L = 4,500 \text{ \mu m}/20 \text{ \mu m}$). As shown in Supplementary Fig. 31, 32, transfer characteristics curve displays a similar hysteresis under luminescence, indicating charge trapping-and-detrapping process similar to the Ag NCs/pentacene device. The C8-BTBT based device displays a 10^4 dynamic range of current levels. However, the overall performance of the device is inferior to that of pentacene-based device.

On Page S42, we have appended:

Supplementary Fig. 31 | Transfer characteristics of C8-BTBT-based device. Hysteric transfer characteristics window generates under white light (10 mW cm^{-2}).

Supplementary Fig. 32 | I_{DS} retention within a timescale of 2,000 s of C8-BTBT-based device.

Comments 4:

g_{CD} value for nanoclusters is relative lower than other reported chiral materials. Why ?

Response:

We thank the reviewer for this comment and we have also noticed this result. There is a tradeoff between g_{CD} value and roughness of the nanoclusters film. To pursue a uniform and homogeneous interface between nanoclusters and semiconductors, we have to suppress the crystallization and size of nanoclusters. This inevitably resulted in a low g_{CD} value. In another aspect, as chiral materials, bulk materials and nanowires are usually reported with significantly high g_{CD} values due to their regular structures. However, due to the random assembly and aggregation of nanoclusters, their g_{CD} values are offset and kept at a scale of 10^{-3} .

Comments and Responses for Reviewer #2.

Comments 1:

Q1. In the proposed ACP device, the synergistic modulation effect of gate bias and light stimulation is of paramount importance in influencing the photo response. Do mantis shrimps also demonstrate variations in their photo response in accordance to external stimuli other than light? Or is there similar situation in R8 that the capacity of Ag NCs in ACP device is modulated by external stimuli?

Response:

We thank the reviewer for drawing our attention to this: according to the literatures as we find (Nature 411, 547–548 (2001)), mantis shrimps can modulate their photo response behaviors by external stimuli. Their eyes adapt to various circumstances and response to external stimuli. They can tune the filters (transparent, colored filters are placed in front of receptors) according to the changes in water depths, as shown below. These mantis shrimps express an impressive degree of phenotypic plasticity in tailoring their visual systems to their habitats.

(The figure is from the literature cited as ‘Cronin, T., Caldwell, R. & Marshall, J. Tunable colour vision in a mantis shrimp. Nature 411, 547–548 (2001). <https://doi.org/10.1038/35079184>’)

[Reproduced with permission from SNCSC.]

Unfortunately, R8 cell variations under external stimuli have not been reported to our knowledge.

Comments 2:

Q2. In Fig. 4c, the case of $V_g=0V$ with light increases the amount of the current due to the hole accumulation in the pentacene channel. Then, why does the current decrease in the case of negative V_g ? The gate bias seems to only control a dynamics charge trapping of photo-generated carriers into Ag NCs, not the polarity of photo-created carriers.

Response:

Thank you for your comments. For the question about current variation under bias voltage, we give explanation as following: In Fig. 2d, counterclockwise hysteresis is shown in blue line under luminescence. Current increases in the regime where the $V_g = 0V$ or $V_g > 0V$, while current decreases in the reverse sweeping regime where $V_g < 0V$. In this case, V_g is employed to tune the carrier concentration levels in the channel. According to the light valve model and the charge reservoir model, the upshift-and-downshift direction of the current under illumination only depends on the carrier concentration level in the channel before illumination. In other words, when the channel is depleted with application of V_g , drain currents will increase the instant light is turned on. Inversely, attenuation of drain current will take place when gate induced accumulative channel and light on are simultaneously satisfied. Therefore, when a negative V_g is given, ACP is kept in its dormant state with high hole concentration in the channel under darkness. When light is on, electrons in nanoclusters will recombine with a large number of holes in the channel to bring the drain current to a low level. If we use positive V_g and light synergistic modulation at the beginning to accumulate holes in the channel, then under luminescence with $V_g = 0V$, the current will still decrease. Moreover, we agree with the reviewer that gate bias doesn't control the polarity of photo-created carriers and you will find the corresponding revision marked in red **in the manuscript on Page 7** (In the section named 'Charge dynamics in nanocluster-conjugated molecule interface').

'Thus, filled hole/electron levels in pentacene/Ag NCs determine the photoadaptation dynamics. It is worth emphasizing that gate voltage is responsible for charge trapping/detrapping dynamics rather than the polarity of photo-generated carriers in Ag NCs. To conclude, we can facilely modulate ACP through light and gate impulses.'

Comments 3:

Q3. What is the definition of adaptation function? Please describe the adaptation in both cases of the shrimp and in the proposed device.

Response:

Thank you for your suggestive comments. For mantis shrimps, eyes adapt to the variation of ambient light for obtaining the high sensitivity to the light. When mantis shrimps adapt to the dark, the layer of the crystalline cones shortens and the retina becomes longer; the opposite occurs in light

adaptation. The aim of adaptation is to modulate the perception range of eyes to attain best contrast and clear images. Adaptation is a similar process that the camera adjusts the aperture to obtain the appropriate amount of photons to prevent overexposure or underexposure of the images.

For our device, photoadaptation function as a reservoir to regulate the channel current, keeping it at an appropriate level. When current level of ACP is low, ACP is sensitive to weak light, photo-generated holes accumulate under luminescence. Once ACP has been exposed for long time, hole concentration in channel reaches up to a high level. Current will decrease under luminescence to adapt to bright luminance. Hence, our device can imitate photoadaptation behaviors of mantis shrimps.

On Page 6-7, we have made corresponding revision marked in red (In the section named ‘Spectral-dependent visual adaptation in artificial photoreceptors’):

‘In response to the changes of environmental brightness, visual adaptation is evolved as a self-regulated activity in various living creatures. The aim of adaptation is to modulate the perception range of eyes to attain best contrast and clear images. Magnitude of visual adaptation can be modulated by both incident wavelengths and intensities. As far as ACP, photoadaptation function as a reservoir to regulate the channel current, keeping it at an appropriate level. ACP shows stepwise strengthened, electric-mode adaptation with increased V_G (Fig. 4a).’

Comments 4:

Q4. The chiral structure of Ag NCs plays a crucial role in ACP devices, but no peaks were detected in both TEM and XRD results. An evidence (ex. XRD, or TEM diffraction pattern) is needed to confirm the chiral Ag NC crystal structure.

Response:

Thank you for your comments. In ACP device, crystallization of Ag NCs has been deliberately suppressed to obtain a flat and low-defect interface between NCs/organic semiconductor. In fact, we have already obtained the structure of Ag NCs crystal according to the method. The chiral structures of Ag NCs have been displayed in Fig. 1d and Supplementary Fig. 16. The structures of Ag NCs have been included by Cambridge Crystallographic Data Centre (CCDC). You can find all structures of nanoclusters in CCDC, by searching doi: 10.1126/sciadv.aay0107.

On Page S6, supplementary experiment has been performed and appended in the **Supplementary Fig. 5**. We control the process of crystallization of Ag NCs film by regulating the spin-coating rate

and solvents. We obtain XRD results in various conditions.

Supplementary Fig. 5 | XRD of the Ag NCs film. The blue line: Ag NCs film is prepared by high-speed spin-coating method mentioned in manuscript to suppress the crystallization process. The orange line: Ag NCs films is formed by dipping volatile Ag NCs solution on substrate.

We revise the first paragraph in the section named ‘Microstructure and in-sensor charge reservoir’ **in the main text on Page 3-4** as follows. ‘It is discovered that given sufficient nucleation and self-assembly time, Ag NCs form highly crystalline aggregates (Supplementary Fig. 5). However, it represents a challenge for the fabrication of uniform interface with organic semiconductors. We carefully control crystallization and aggregation of Ag NCs by high-speed spin-coating with volatile solvents, hence an ultrasmooth film with a root-mean-square roughness of 0.45 nm is obtained (Supplementary Note 4). No significant X-Ray diffraction (XRD) peaks is observed in as-fabricated Ag NCs layer, being adequate for homogeneous interface formation in conjunction with pentacene (Supplementary Fig. 6 and 7). Crystallinity of pentacene deposited on Ag NCs is effectively suppressed with respect to that on silica (Supplementary Fig. 7).’

Comments 5:

Q5. When evaluating the electrical characteristics of a transistor with the channel exposed, it is highly susceptible to external environmental influences. Thus, it is needed a comment on the measurement conditions. For instance, were the measurements taken in a vacuum or in ambient atmospheric conditions? Are there any differences in the retention results depending on the environment?

Response:

Thank you for your comments. The modifications and additions are attached to **Supplementary information on Page S43**.

Supplementary Note 6: Environmental influences on electrical performance of ACP arrays.

Characteristics curves of our devices have been tested under nitrogen protected circumstance at 25°C and ambient atmosphere with various humidity at 25°C. Although the off currents of the devices have raised up in high humidity, as shown in Supplementary Fig. 33, our devices still enable to operate multifunction in kinds of circumstances. Moreover, our devices still retain good performance after being deposited in low level of oxygen and water circumstance for one year. The devices still keep retention capacity as shown in Supplementary Fig. 16.

On Page S43, we have appended:

**Supplementary Fig. 33 | Transfer characteristic curves of ACP devices exposed to various circumstance.****Comments 6:**

Q6. In Supplementary Fig. 8, examining the hysteresis loop under different wavelength, it is evident that selectivity is insufficient at lower light intensities depending on the RGB wavelength. What accounts for this phenomenon?

Response:

Thank you for your comments. Three main reasons account for insufficiency of spectral selectivity.
(i) Nanoclusters as photosensitive materials compose an 8-nm-thick underlayer, to be buried beneath

an organic semiconductor layer with thickness of 30 nm. Therefore, photogenerated excitons in nanoclusters could be hardly induced by weak light. Photocurrent changes under weak lights with different wavelengths are indistinctive. (ii) Pentacene has a very broad spectral absorption but poor absorption efficiency. Nanoclusters only have strong absorption coefficients for ultraviolet and blue light. Therefore, for red light and green light, the photocurrent generated basically comes from pentacene, thus limiting the selectivity of lights with panchromatic wavelengths. (iii) The sensitivity of our devices to dim light is limited by relative low absorption coefficient and low mobility of materials. The performance of our device under weak light needs to be improved.

Comments 7:

Q7. In Supplementary Fig. 11, under the conditions of $V_G=60$ V and 460 nm illumination at 2 mW cm^{-2} , stable retention was observed even at a timescale of 10,000 s. However, in Supplementary Fig. 13, the multi-level retention only extends to a timescale of 600 s despite the bias duration being raised to 5 s. Why is this the case?

Response:

Thank you for your comments and we have performed supplementary experiments to address your concern. In supplementary Fig.13, we emphasize the capacity of multi-levels but not retention, we just test 600s to demonstrate controllable modulation of our device to keep their current values into multiple intervals. The results do not mean that the retention time can only be maintained for 600s. Hence, the ACP devices were measured in current (I) – time (t) mode to demonstrate multi-level retention for at least 10000s, as shown in **Supplementary Fig. 16 on Page S17**.

Supplementary Fig. 16 | Multi-level retention within a timescale of 10,000 s. Conjoint modulation of V_G pulse and 460 nm (blue) light stimuli enable multi-level retention signals in an ACP.

Comments 8:

Q8. (minor) The full abbreviation of LCPR, RCPR is not mentioned at any point.

Response:

Thank you for your comments. The full abbreviation of LCPR, RCPR are left circularly polarized light and right circularly polarized light, respectively.

We thank you again for your valuable suggestions and comments.

Comments and Responses for Reviewer #3.

Comments 1:

Stomatopods, commonly known as mantis shrimp have one of the most complex visual systems known in the animal kingdom, with 12 spectral receptors and the ability to perceive linearly and circularly polarized light.

I find the work original even if I don't understand everything, especially the materials part, it's hard to follow since it's not my area of expertise.

I haven't found an article close to this present one, by the same authors.

The most related works on this subject are

Altaqui, A., Sen, P., Schrickx, H., Rech, J., Lee, J. W., Escuti, M., ... & Kudenov, M. (2021). Mantis shrimp-inspired organic photodetector for simultaneous hyperspectral and polarimetric imaging. *Science Advances*, 7(10), eabe3196.

<https://www.science.org/doi/full/10.1126/sciadv.abe3196>

Authors doesn't quote it. The materials used are not the same.

I would ask to quote

Haessig, G., Joubert, D., Haque, J., Chen, Y., Milde, M., Delbruck, T., & Gruev, V. (2021). Bio-inspired polarization event camera. arXiv preprint arXiv:2112.01933.

<https://arxiv.org/pdf/2112.01933>

and

Garcia, M., Davis, T., Blair, S., Cui, N., & Gruev, V. (2018). Bioinspired polarization imager with high dynamic range. *Optica*, 5(10), 1240-1246. <https://opg.optica.org/optica/fulltext.cfm?uri=optica-5-10-1240>

and

D. Floreano, R. Pericet-Camara, S. Viollet, F. Ruffier et al. (2013)

Miniature curved artificial compound eyes

Proceedings of National Academy of Sciences of USA, PNAS, 2013 Jun 4, 110(23):9267-72

<https://doi.org/10.1073/pnas.1219068110>

Response:

Thank you for your comments. We are very grateful to the reviewer for helping us find related literatures that we neglect. These literatures are innovative researches on polarization imaging and artificial compound eye inspired from mantis shrimps. In this newly revised manuscript, we have quoted them as shown **in the main text on Page 14** and **in Supplementary information on Page S46**.

References

35. Garcia, M. et al. Bioinspired polarization imager with high dynamic range. *Optica*. 5, 1240-1246 (2018).
36. Floreano, D. et al. Miniature curved artificial compound eyes. *P Natl Acad Sci USA*. 110, 9267-9272 (2013).
37. Haessig, G. et al. Bio-inspired Polarization Event Camera. arXiv:2112.01933 (2021).

Comments 2:

The size of the text in the figures is really small everywhere.

Figure 4c explains their position in relation to the state of the art, but it's very small.

For me, it's the most important figure, but the choice of colors isn't very legible, and the font size isn't correct.

There's a lot in their paper in terms of figures, Authors should move more to the Supp info to lighten the paper and make the figures bigger.

Response:

We thank the reviewer for pointing out our problems on figures. We take your suggestions and we have changed all of the figures with bigger fonts and legible colors both **in the main text and Supplementary information**.

Comments 3:

Most importantly, the authors should relax the biomimetic aspect with respect to a full mantis shrimp eye as the mantis shrimp eye has 12 spectral receptors and the ability to perceive linearly and circularly polarized light.

Response:

Thank you for your valuable comments. Following with your suggestions, we have added

additional explanations to the mantis shrimp vision section in Supplementary information. We consider that complex vision systems of mantis shrimps can hardly be expressed in a few words in the main text. We grab the critical information to illustrate how ACP are designed by inspiration of anatomical structure and functions of mantis shrimps vision systems. To supply more details on the vision systems, we append a supplementary note for complements.

In Page S34-35, we have appended:

Supplementary Note 1: Structure and functions of mantis shrimp eyes

Part 1: Structure

Supplementary Schematic 1 | The structure of mantis shrimp visual system.

Mantis shrimps possess apposition compound eyes with 16 anatomically different photoreceptor types. Each eye can be sub-divided into three regions named dorsal hemispheres (DH), six-row midband (MB) and ventral hemispheres (VH), respectively. MB region with six closely-spaced parallel rows of ommatidia, composed of the corneal lens, the crystalline cone and the rhabdom. The rhabdom consists of a short photoreceptor cell (R8) and seven long cells (R1 - R7), 14 types of these cells are found in MB region. Each cell has interdigitating, coplanar microvilli arranged in specific directions. In rows 2 and 3, colored filters constructed by aforementioned cells situate between rhabdom tiers. Optical nerves connect the end of rhabdom and extend to the brain.

Part 2: Functions

Color vision:

Mantis shrimps perceive the visual information through 12 channels of colors while humans only employ 3 channels. Each type of photoreceptor picks up a specific color, sampling a narrow set of wavelengths ranging from deep ultraviolet to far red (300 to 720 nm). R8 cell is sensitive to violet spectral region while R1 to R7 cells mainly response to 400-700 nm part of the spectrum.

Adaptative vision:

Mantis shrimps tunes shapes of crystalline cones and rhabdoms to execute dark/light adaptation. Cones shorten while rhabdoms extend during dark adaptation. Opposite behavior occurs in light adaptation. Photomechanical adaptation is a powerful strategy to change the amount of photons accepted by eyes, which plays a similar role as the aperture of cameras or the pupil of eyes. Moreover, color filters are affected by depth of water, switching light absorption range to adapt to low-level light environment.

Circularly polarization vision:

Considering the microvillar structure of photoreceptor cells, R8 cell act as 1/4 wave plates with a fast axis parallel to their microvillar planes. R8 cell enables the conversion of circularly polarized light into linearly polarized light, then the signal is transferred and processed by the R1-R7 cells. In rows 5 and 6, because of an angle of 45° orientation of the R8 and R1-R7 microvillar planes, the R1-R7 cells suffice to distinguish clockwise-rotating and counterclockwise-rotating electric vectors.

Comments 4:

I cannot see a comparison in terms of optical characteristics with the Mantis Shrimp natural counterpart eye, as done in the table 1 of Floreano et al. PNAS 2023

Response:

Thank you for your comments. We also supplement a table based on the data of our devices and natural eyes for comparison to more clearly demonstrate the capabilities of our devices.

On Page 16, main text, we have appended:

	ACP arrays	Mantis shrimp vision system
Spectral range	UV, RGB	UV, RGB
Layer number	2	8
Thickness	40 nm	640 μm
Adaptation time	0.45 s	Hours
Perception range	120	NA
Memory retention time	Over 10,000 s	NA
Sites of color vision	nanoclusters; pentacene	R1 - R8; filters
Photoadaptation mechanism	nanocluster-conjugated molecule interface	filters, photomechanical changes
Sites of CPL perception	nanoclusters	R8

Table 1 | A comprehensive comparison of ACP arrays and mantis shrimp vision system.

Comments 5:

Finally, it would have been nice to see a picture or a conceptual view of the final device that would include the electronic read out. I would be very helpful.

Response:

Thank you for your comments. We have done a conceptual schematic for our device and we give detailed notes attached to the schematic to present a clear framework of our devices and arrays.

On Page S37-38, we have appended:

Supplementary Schematic 2 | The circuit diagram of ACP arrays to realize (a) color vision, (b) adaptive vision and (c) circular polarization vision.

Supplementary Note 3: All-in-one system of biomimetic ACP

Although a few mantis-shrimp-inspired artificial photoreceptors have been fabricated for polarimetric imaging and high dynamic range machine vision⁹⁻¹¹, none of them have noticed the significance of multi-functional integration. All-in-one system of ACP is learned from mantis shrimps that color vision, adaptative vision and circular polarization vision are integrated into a single unit. It is known to us that multifunctional visions of mantis shrimps require various photoreceptor cells, color filters and photomechanical changes of crystalline cones. Biomimetic ACP process these visions just by synergistic modulation of V_G and light on nanocluster-conjugated molecule interface as summarized in Table 1. For color vision, difference of absorption range of nanoclusters and pentacene endow ACP with distinguished response to spectral lights. For adaptative vision, nanoclusters function as light-response charge reservoir to control account of charge carriers in channel. Under the influence of nanoclusters, the channel current tends to remain in an equilibrium state. In other words, high or low charge carrier density in channel will eventually return to the intermediate value under illumination. For circular polarization vision, chirality of nanoclusters endows ACP to recognize the tiny difference of left/right circularly polarized lights. Because of charge trapping and detrapping capacity of nanoclusters, ACP even successfully achieve CPL memory and CPL adaptation.

REVIEWERS' COMMENTS

Reviewer #1 (Remarks to the Author):

The authors have convincingly addressed the points raised during the first round. The paper is now ready for being accepted in Nat Comm.

Reviewer #2 (Remarks to the Author):

All my concerns are addressed in the revised manuscripts. Thanks.

Reviewer #3 (Remarks to the Author):

Nice revision.

I have no further comments.